



# Surface global and diffuse solar radiation over China acquired from geostationary Multi-functional Transport Satellite data

Hou Jiang[1,2], Ning Lu[1,4], Jun Qin[3], Ling Yao[1,4]

[1]State Key Laboratory of Resources and Environmental Information System, Institute of Geographic Sciences and Natural Resources Research, Chinese Academy of Sciences, Beijing, 100101, China
[2]College of Resources and Environment, University of Chinese Academy of Sciences, Beijing, 100190, China
[3]Key Laboratory of Tibetan Environment Changes and Land Surface Processes, Institute of Tibetan Plateau Research, Chinese Academy of Sciences, Beijing, 100085, China
[4]Jiangsu Center for Collaborative Innovation in Geographical Information Resource Development and Application, Nanjing, 210023, China

*Correspondence to*: Ning Lu (lvn@lreis.ac.cn)

**Abstract.** Surface solar radiation drives the water cycle and energy exchange on the earth's surface, being an indispensable parameter for many numerical models to estimate soil moisture, evapotranspiration and plant photosynthesis, and its diffuse component can promote carbon uptake in ecosystems as a result of improvements of plant productivity by enhancing canopy light use efficiency. To reproduce the spatial distribution and spatiotemporal variations of solar radiation over China, we generate the high-accuracy radiation datasets, including global solar radiation (GSR) and the diffuse radiation (DIF) with spatial resolution of 1/20 degree, based on the observations from the China Meteorology Administration (CMA) and Multi-functional Transport Satellite (MTSAT) satellite data, after tackling the integration of spatial pattern and the simulation of complex radiation transfer that the existing algorithms puzzle about by means of the combination of convolutional neural network (CNN) and multi-layer perceptron (MLP). All data cover a period from 2007 to 2018 in hourly, daily total and monthly total scales. The validation in 2008 shows that the root mean square error (RMSE) between our datasets and in-situ measurements approximates 73.79 W/m$^2$ (0.27 MJ/m$^2$) and 58.22 W/m$^2$ (0.21 MJ/m$^2$) for GSR and DIF, respectively. Besides, the spatially continuous hourly estimates properly reflect the regional differences and restore the diurnal cycles of solar radiation in fine scales. Such accurate knowledge is useful for the prediction of agricultural yield, carbon dynamics of terrestrial ecosystems, research on regional climate changes, and site selection of solar power plants etc. The datasets are freely available from Pangaea at https://doi.org/10.1594/PANGAEA.904135 (Jiang and Lu, 2019).

## 1 Introduction

Solar radiation is the main energy source for life on Earth (Wild, 2009), the major driving forces of the climate system (Mueller et al. 2009) and a key factor affecting surface-radiation budget (Liang et al., 2010). In recent decades, research on quantitative estimation of global solar radiation (GSR) and the proportion of diffuse components has attracted growing



interest in view of its great scientific value and socioeconomic benefits (Greuell et al., 2013). GSR is a prerequisite for modelling terrestrial ecosystem productivity (Jacovides et al., 2007), and an essential element for estimating heat fluxes, soil moisture and evapotranspiration etc. (Zhang et al., 2004). The distribution and intensity of GSR are indispensable for site selection of solar photovoltaic power and further estimation of its production (Pravalie et al., 2019). Previous studies reveal

that the diffuse solar radiation (DIF) contributes to the ecosystem carbon uptake as it can reduce photosynthetic saturation and increase the canopy light use efficiency (Kanniah et al., 2012; Mercado et al., 2009). To simulate the carbon dynamics of terrestrial ecosystems, the explicit knowledge on DIF is urgently required by the ecological models (Alton et al., 2007). The effects of diffuse radiation on plant productivity have also become a popular issue in the field of ecology and environmental sciences (Gu et al., 2002; Mercado et al., 2009; Zhang et al., 2011; Zhang et al., 2017).

Interpolation/extrapolation of direct measurements from radiation stations is a traditional way to obtain continuous surface solar radiation data. Although great efforts have made to establish globally covered surface-radiation measurement networks, such as NOAA's Surface Radiation budget network (SURFRAD), the Baseline Surface Radiation Network (BSRN), the World Radiation Data Centre (WRDC) and the Global Energy Balance Archive (GEBA), it remains insufficient to derive the high-precision radiation distribution from surface observations alone because of the sparsity and heterogeneity

of stations (Liang et al., 2006). Due to the intrinsic limitation that interpolation/extrapolation is unable to reproduce variability smaller than twice the average distance between stations, large inaccuracies usually appear because GSR and DIF are largely variable in both time and space (Beckers, 2012). Since meteorological variables are commonly available and easily accessible, empirical models such as temperature-based models, sunshine duration-based models, and relative humidity- and cloud-based models are developed to reproduce the spatial distribution of surface radiation (Besharat et al.,

2013; Jacovides et al., 2007), but their accuracies are strongly affected by the deficient stations under neglected maintenance and/or insufficient calibration schedule (Dumas et al., 2015). Retrieval from satellite observations is another reliable approach to gain spatially continuous estimates of surface radiation as the digital signals received by sensors carry massive information about the atmospheric state and underlying land surface (Qin et al., 2015). These algorithms mainly include two categories: constructing empirical mathematical relationships between top of atmosphere (TOA) and surface radiative fluxes

(Linares-Rodriguez et al., 2013; Lu et al., 2011) and driving complex radiative transfer models utilizing satellite-derived atmospheric parameters (Greuell et al., 2013; Huang et al., 2011). However, the above methods depend on an independent pixel approximation which takes no consideration of spatial collocation of surface radiation (Oreopoulos et al., 2000). Large biases and uncertainties are likely to occur in situations under complex broken clouds when comparing to quality-controlled ground observations (Wyser et al., 2005). Besides, due to the complexity of integrating spatial pattern and simulating

radiation transfer processes, significant inaccuracies persist in the parameterization of clouds and their interaction with other processes (Rasp et al., 2018).

To tackle the aforementioned shortcomings, especially the independent pixel approximation caused, we introduce a hybrid deep network to retrieve hourly GSR/DIF from geostationary satellite data (Jiang et al., 2019). The key idea is



utilizing convolutional neural network (CNN) to extract spatial pattern from satellite imagery and then linking the abstract patterns as well as time/location attributes to target hourly radiation values through multi-layer perceptron (MLP). The representative characteristic lies in its ability to characterize changeable cloud morphology and approximate arbitrary complicated non-linear functions. After complete learning and optimization, the trained network can easily be adapted to

5    new surface patterns and provide fast predictions during inference time. Based on such a novel network, we generate the high-accuracy radiation datasets over China from 2008 to 2018, including GSR and DIF with a spatial resolution of 1/20 degree, reproducing the spatial distribution and diurnal/seasonal variations of solar radiation in fine scales. All datasets have been published in Pangaea (https://www.pangaea.de/), to support radiation-related applications and scientific researches particularly on regional climate change and utilization of renewable energy. In this paper, we describe the structure of our

10   deep network, workflow of the radiation data production as well as the validation accuracy of our products. Spatiotemporal deviations of hourly GSR and DIF at different stations are presented, and spatially continuous maps of our predictions are compared to other widely used products. Such analysis and comparison are very helpful for rational utilization of the products, in-depth understanding of error causes, and potential improvements of next generations.

## 2 Data and methods

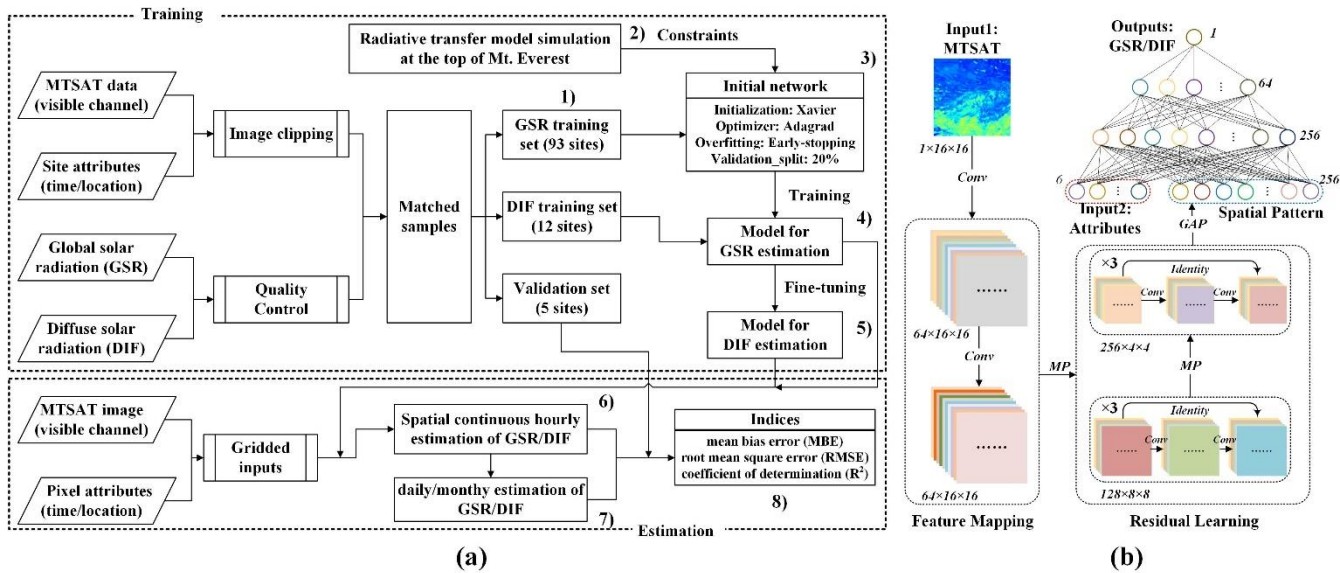

**Figure 1: The algorithm used for radiation data production. (a) The flowchart to generate high-resolution GSR and DIF; (b) The structure of the hybrid deep network.** *Conv* **represents convolution operation;** *MP* **means max-pooling operation; and** *GAP* **stands for global average-pooling operation. The size of 3D blocks are labelled below as channels × width × height. ×3 means that the sequential operations in the block are repeated three times. Numbers 1-8 correspond to the main procedures listed in section 2.3.**





The schematic workflow of the radiation datasets production is shown in figure 1a, with a detailed structure of the initial network displayed in figure 1b. The deep network takes Multi-functional Transport Satellite (MTSAT) satellite image blocks as well as the local time (month, day, and hour) and location (longitude, latitude and altitude) attributes as inputs, and outputs GSR or DIF values. The entire process for data production consists of two main sections: training and estimation.

The training section concentrates on learning the underlying non-linear relationships between inputs and target outputs, while the estimation section is responsible for predicting spatially continuous GSR and DIF maps using the optimized networks by feeding gridding inputs. The details are described in the following.

## 2.1 Basic data

The data used here include hourly GSR, hourly DIF, geostationary satellite images, as well as digital elevation model (DEM)

data. The radiation measurements along with the time/location attributes of involved stations are from China Meteorological Administration (CMA) (http://data.cma.cn/ last accessed: 16 Oct., 2019), covering a period from 1 Jan., 2008 to 31 Jan., 2008. Figure 2 shows the spatial distribution of all stations, of which 81 sites (red circles) only provide GSR while the rest (blue triangles) provide both GSR and DIF. The elevations of these stations vary from 2.5m to 4507m, with different climates and various land cover types including desert, semi-desert, wetlands, croplands, grasslands and forests etc., making

it possible to collect enough representative samples for deep network training. Specially, five stations (the labelled triangles) with the same underlying surface but different climatic characteristics are picked out for independent validation of the spatial extendibility of our deep network. Further quality checks are made according to the physical threshold test (Roebeling et al., 2004), i.e., an upper limit of 0.9 for the atmospheric clearness calculated as the hourly measured GSR relative to the hourly potential extra-terrestrial GSR on a horizontal surface. Such test eliminates most of the wrong samples with quite few

escaped ones which makes no sense seeing that the robustness of deep network allows the existence of few errors. In total, 0.49% of all records not passing the test are deleted, and 441547 samples for GSR and 55096 samples for DIF are retained.





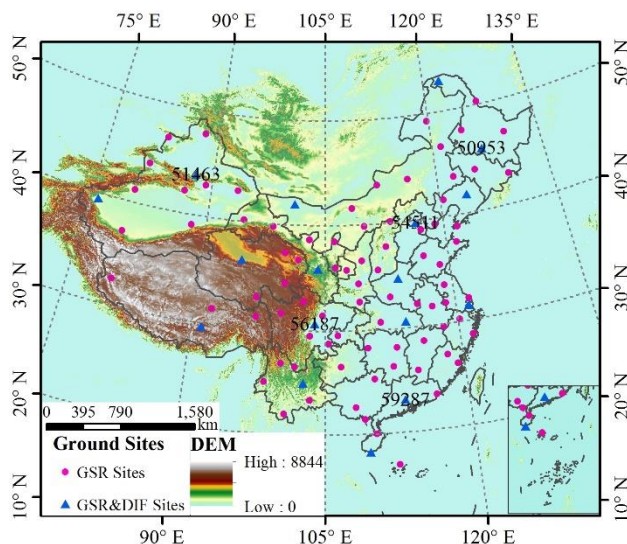

**Figure 2: Locations of radiation stations used in our study. DEM data from Shuttle Radar Topography Mission represents the surface elevation in meters. Sites with labels are used for model validation while others are for model training or fine-tuning.**

The used geostationary satellite images are Multi-functional Transport Satellites (MTSAT) data, provided by the Japan
Meteorological Agency (JMA). The MTSAT-1R, launched in February 2005, is positioned at 104 °E above the equator to monitor Asia-Pacific region (70 °N - 20 °S, 70 °E - 160 °E). The imager on board MTSAT scans the surface every 30 min and provides images in five channels: one visible channel (VIS, 0.55-0.80μm), two split-window channels (IR1, 10.3-11.3μm; IR2, 11.5-12.5μm), one water vapour channel (IR3, 6.5-7.0μm) and one shortwave infrared channel (IR4, 3.5-4.0μm). The spatial resolution of MTSAT images is 1 km for the visible channel, and 4 km for the other infrared channels. Our used
MTSAT products are available at http://weather.is.kochi-u.ac.jp/ (last accessed: 16 Oct., 2019). DEM data are from Shuttle Radar Topography Mission that generates the most complete high-resolution digital topographic database of the Earth, covering over 80% of the Earth's land surface between 60 °N and 56 °S with data points posted approximately 30m. The data can be obtained from the website http://srtm.csi.cgiar.org/SELECTION/inputCoord.asp (last accessed: 16 Oct., 2019).

## 2.2 Deep network

Deep learning has achieved notable success in modelling ordered sequences and data with spatial context in the field of computer vison, speech recognition and control systems (LeCun et al., 2015). In geosciences, convolutional neural networks (CNN) are widely utilized to extract spatial features for definition and classification of extreme situations, for instance, storms, spiral hurricanes, and atmospheric rivers (Reichstein et al., 2019). Extending on previous classical neural networks (He et al., 2016; Simonyan and Zisserman, 2014), herein we build a hybrid deep network for radiation prediction from
geostationary satellite data. The structure is shown in figure 1b and the detailed configurations are listed in table 1. There are two input pipes: *Input1* for MTSAT image blocks and *Input2* for attributes. The designed size of input blocks is $1 \times 16 \times 16$





pixels (visible channel × width × height), equalling around 80km × 80km on the ground. Firstly, the MTSAT images are mapped into multidimensional features through two sequential convolutional layers (*Conv*) with 64 kernels, followed by a max-pooling (*MP*) with 2 × 2 kernel, stride of 2. Then residual blocks (He et al., 2016), equipped with 128 kernels and 256 kernels, respectively, are utilized in sequence to extract abstract spatial pattern implicit in satellite imagery. These sequential

operations are repeated three times and another MP layer with 2 × 2 kernel, stride of 2 is embedded between them for context aggregation. The resulting 256 × 4 × 4 feature maps are flattened into ordinary vector by a global average-pooling (*GAP*) layer before flowing into the prognostic multi-layer perceptron (MLP) together with additional attributes, normalized information on the local time (month, day and hour) and location (longitude, latitude and altitude) of ground stations corresponding to the central point of MTSAT block. The two hidden fully-connected layers (*FC*) of MLP are implemented

with 256 and 64 nodes, respectively, to simulate the nonlinear relationships between inputs and outputs, i.e., GSR or DIF. It is stressed that above convolutional layers are followed by the rectified linear unit (*ReLU*) (Nair and E. Hinton, 2010) activation function and batch normalization (Ioffe and Szegedy, 2015) layers to speed up the learning process and alleviate the repulsive vanishing gradient problems, and all convolution kernels are of size 3 × 3, the smallest kernel to seize spatial patterns in different directions. Such a network is expected to resolve the integration of spatial pattern and simulation of

complex radiation transfer, thus further improve the accuracy of radiation inversion from satellite images.

**Table 1.** Detailed configurations of our deep network. *Conv*: convolutional layer; *MP*: max-pooling layer; *RB*: residual block; *GAP*: global average-pooling layer; *FC*: fully-connected layer; MLP: multi-layer perceptron; *ReLU*: rectified linear unit.

| Module | Unit | Input Size | Kernel Num. | Kernel Size | Stride | Activation Function | Output Size |
|---|---|---|---|---|---|---|---|
| | *Input1* | - | - | - | - | - | 1 × 16 × 16 |
| Feature Mapping | *Conv* | 1 × 16 × 16 | 64 | 3 × 3 | 1 | *ReLU* | 64 × 16 × 16 |
| | *Conv* | 64 × 16 × 16 | 64 | 3 × 3 | 1 | *ReLU* | 64 × 16 × 16 |
| | *MP* | 64 × 16 × 16 | - | 2 × 2 | 2 | - | 64 × 8 × 8 |
| Residual Learning | *RB* | 64 × 8 × 8 | 128 | 3 × 3 | 1 | *ReLU* | 128 × 8 × 8 |
| | *RB* | 128 × 8 × 8 | 128 | 3 × 3 | 1 | *ReLU* | 128 × 8 × 8 |
| | *RB* | 128 × 8 × 8 | 128 | 3 × 3 | 1 | *ReLU* | 128 × 8 × 8 |
| | *MP* | 128 × 8 × 8 | - | 2 × 2 | 2 | - | 128 × 4 × 4 |
| | *RB* | 128 × 4 × 4 | 256 | 3 × 3 | 1 | *ReLU* | 256 × 4 × 4 |
| | *RB* | 256 × 4 × 4 | 256 | 3 × 3 | 1 | *ReLU* | 256 × 4 × 4 |
| | *RB* | 256 × 4 × 4 | 256 | 3 × 3 | 1 | *ReLU* | 256 × 4 × 4 |





|  | GAP | 256 ×4 ×4 | - | 4 ×4 | 0 | - | 256 |
|---|---|---|---|---|---|---|---|
| | Input2 | - | - | - | - | - | 6 |
| MLP | FC | 262 | - | - | - | ReLU | 256 |
| | FC | 256 | - | - | - | ReLU | 64 |
| | FC | 64 | - | - | - | ReLU | 1 |

## 2.3 Dataset generation

Figure 1a illustrates the explicit process to generate data production process from MTSAT images. The main procedures are as follows:

1) Prepare the training datasets. For each ground station in 2008, its corresponding 16 ×16 neighbouring pixels are cut out from MTSAT images and matched with quality-controlled GSR/DIF records in 2008 according the time information. These samples are then separated into three groups: GSR training set (93 symbols in figure 2 without labels), DIF training set (12 triangles in figure 2 without labels) and validation set (5 triangles in figure 2 with labels).

2) Simulate the state at the top of Mt. Everest. To guarantee that in high elevation regions the final trained network has a reasonable extrapolation, constraints from radiative transfer model simulation at the top of Mt. Everest are mixed into the GSR/DIF training set. The Santa Barbara DISORT Atmospheric Radiative Transfer (SBDART) model is adopted for the simulation (Lu et al., 2011).

3) Initialize the deep network. The network is implemented using the *keras* (Chollet and others, 2015) package with the *theano* backend (Theano Development Team et al., 2016). All parameters of the network are initialized through Xavier (Glorot and Bengio, 2010). The validation split is set to 20%, meaning that 80% of the training samples are randomly selected to determine the network parameters while the remaining samples serve as validation ones to identify whether the network is overfitting during training section. The learning rate is initially 0.01 but multiplied by 0.5 across a learning plateau.

4) Train the network for GSR prediction. After initialization, the Adagrad optimizer (Duchi et al., 2011) is used to iteratively find the optimal weights and biases that minimize the mean-squared error (MSE) between the network's predictions and the training targets. An early-stopping mechanism which relinquishes on further optimization when the performance ceases to improve sufficiently or even degrades, is utilized to relieve overfitting by monitoring the network's performance on the validation part, randomly selected 20% of the GSR training set. The model with the best performance is preserved for subsequent predictions, i.e., gaining spatially continuous radiation maps or estimates for other years.

5) Fine-tune the network for DIF prediction. The gained knowledge of the trained network in 4) is from GSR labels through supervised learning, making it not completely suitable for DIF estimation. Thus, DIF training set is used to fine-





tune the network through the transfer learning technique which gets new skills while recycling acquired knowledge. The best model can be obtained in short time as all initial parameters have been optimized once in 4).

6) Generate spatially continuous hourly estimation of GSR/DIF. Hourly MTSAT images are gridded and matched with corresponding pixel attributes on time and location to meet the input criteria. The best models gained in 4) and 5) are

then used to predict GSR and DIF maps, respectively. In addition, the direct solar radiation (DIR) can be derived by subtracting DIF from GSR.

7) Integrate daily and monthly estimation of GSR/DIF. Daily total values are integrated from the predicted hourly maps by adopting the same method during the ground measurements. Monthly total values are the sum of all daily values within the corresponding month.

8) Validate the radiation datasets. The validation set representing ground measurements from 5 independent sites is used for our product evaluation. The overall accuracy is measured using four indices: coefficient of determination ($R^2$, the confidence level is 95%), mean bias error (MBE), root mean-squared error (RMSE), and relative root mean-squared error (rRMSE), between predictions and ground measurements:

$$R^2 = 1 - \sum_{i=0}^{n}(\hat{y}_i - y_i)^2 / \sum_{i=0}^{n}(y_i - \bar{y})^2 , \qquad (1)$$

$$\text{MBE} = \frac{1}{n}\sum_{i=0}^{n}(\hat{y}_i - y_i) , \qquad (2)$$

$$\text{RMSE} = \sqrt{\frac{1}{n}\sum_{i=0}^{n}(\hat{y}_i - y_i)^2} , \qquad (3)$$

$$\text{rRMSE} = 100 \times \text{RMSE}/\bar{y}, \qquad (4)$$

where $n$ is the total number of data samples indexed by $i$, $y$ represents the measured value whose mean value is $\bar{y}$, and $\hat{y}$ is the prediction value from our production.

## 3 Description of the radiation datasets

Our datasets provide the gridded surface global and diffuse solar radiation within 71.025 °E - 141.025 °E and 14.975 °N - 59.975 °N (mainly in China) with an increment of 0.05 °, covering the period from 2007 to 2018. Both the direct predicted hourly values and the integrated daily and monthly total values are available. The dataset should be useful for the analysis of the regional differences and temporal cycles of solar radiation in fine scales, as well as the impact of diffuse radiation on

plant growth etc.

Figure 3 shows an example of the hourly estimation of GSR and DIF at UTC 6:00, 22 Jun., 2008 (BJT 14:00, 22 Jun., 2008). The influence of cloud depth, surface topography and elevation are reflected in the spatial distribution of surface radiation. Under the thick clouds (red regions in figure 3a), both GSR and DIF are lower than surrounding areas. In contrast, with respect to regions below thin clouds (yellow regions in figure 3a), as more DIF is obtained on the surface, GSR is relatively





higher. As for areas under clear sky conditions (blue regions in figure 3a), the surface radiation is relatively stronger in high altitude areas, for example, the Tibetan Plateau.

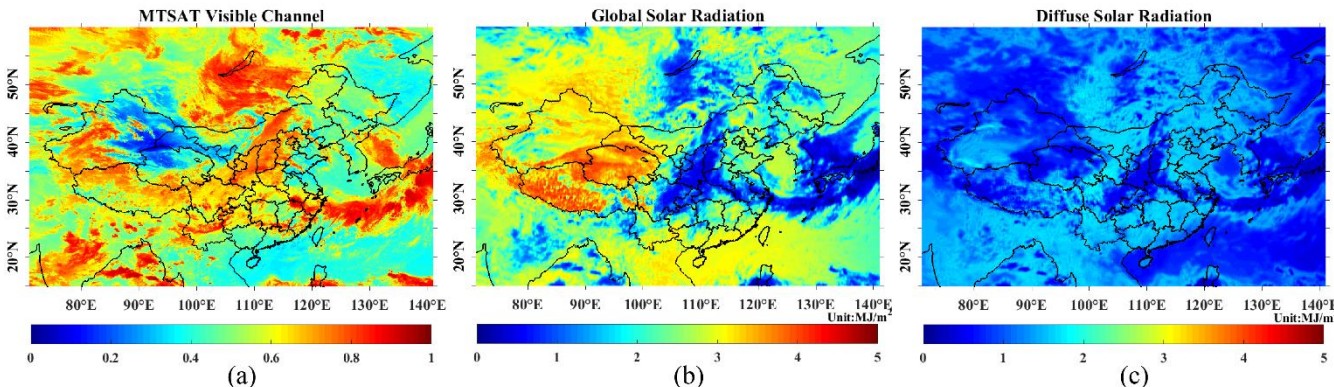

**Figure 3: Spatial distribution of hourly GSR and DIF at UTC 6:00, 22 Jun., 2008 (BJT 14:00, 22 Jun., 2008). (a) Reflectance of MTSAT visible channel; (b) GSR; (c) DIF. The unit of radiation is MJ m$^{-2}$.**



**Figure 4: Spatial distribution of daily, monthly and yearly GSR, DIF and DIR. (a-c) Daily total GSR, DIF and DIR on 22 Jun., 2008; (d-f) Monthly total GSR, DIF and DIR in June, 2008; (c) Yearly total GSR, DIF and DIR in 2008. The legend for each row is labelled at the right. The unit of radiation is MJ m$^{-2}$.**

Figure 4 shows the spatial distribution of total GSR, DIF and the derived DIR in daily, monthly and yearly scales. The daily total radiation on 22 June, 2008 shares the similar characteristics with the hourly radiation in figure 3, indicating a stable atmospheric state and nonviolent weather conditions in the day. At monthly total and yearly total scales, the apparent regional differences are exposed thoroughly. The distribution of solar radiation exhibits obvious latitudinal dependency, but also affected by the surface topography, regional climate and distance to coastal line. Nationwide, GSR is highest on the Tibetan Plateau and the lowest in the Szechwan, while DIF has the maximum value on the North China Plain and the minimum value on the Tibetan Plateau. The direct radiation is predominant in regions with high altitudes (the Tibetan Plateau) or drought climate types (the Mongolia Plateau). DIF is the main source of solar radiation on the surface for areas with abundant rainfall or frequently covered by clouds (the middle and lower reaches of the Yangtze River, the Szechwan





Basin and Guizhou). Although the networks used for prediction are based on the samples in China, it provides satisfying estimation in surrounding areas, for example, in June DIF contributes to the majority of surface radiation in India and Southeast Asia due to the coming of rainy season.

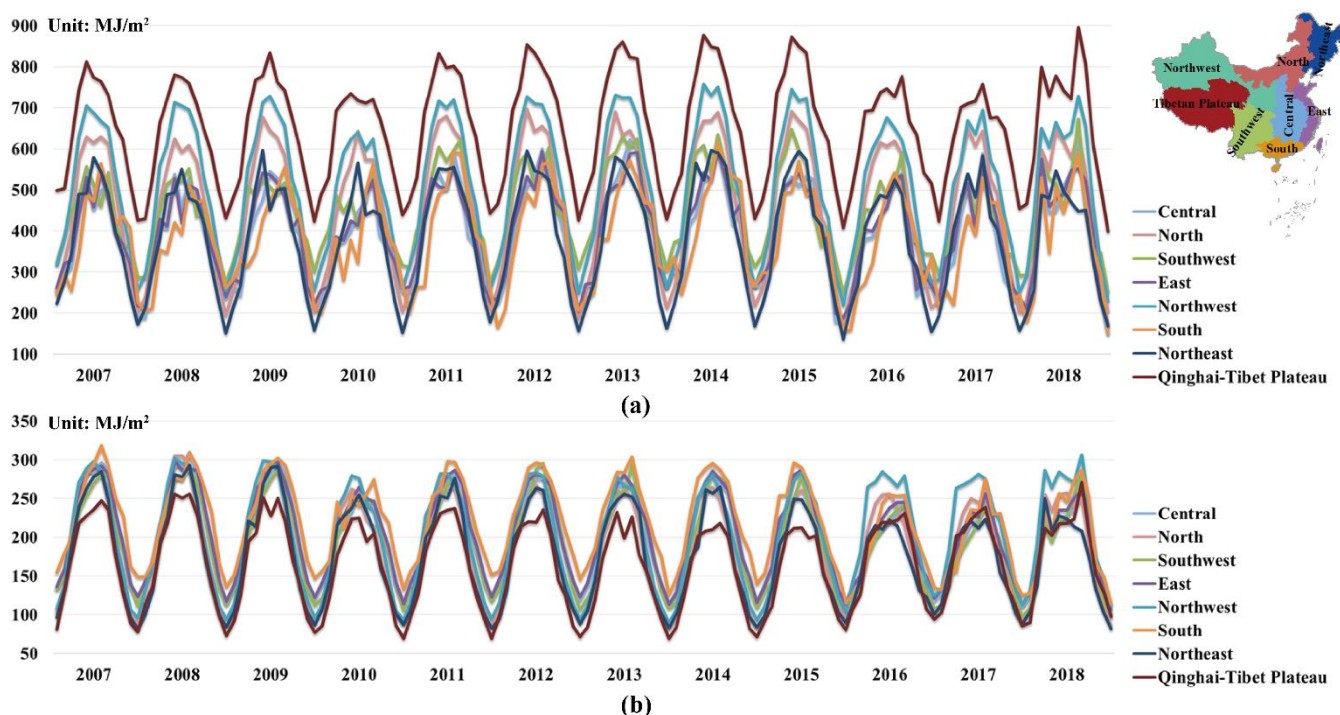

5 **Figure 5: Monthly variations of averaged GSR (a) and DIF (b) in different regions from 2007 to 2018. The values are from the monthly total radiation data in the unit is MJ m⁻². Different regions are as shown at the upper right corner.**

We also establish time series products to observe the temporal variations of surface solar radiation. Figure 5 shows the monthly variations of statistically averaged GSR and DIF for different regions in China from 2007 to 2018. It presents that GSR in the Qinghai-Tibet Plateau is the highest all the year round, owning to its large amounts of direct radiations benefiting 10 from the significantly higher altitudes, which in contrast leads to its lowest received DIF as shown in figure 5b. The proportion of diffuse radiation exhibits the highest in the south of China (relatively lower GSR but higher DIF) compared with other regions due to the frequent cloudy and rainy weather. A slight dimming of GSR is observed in 2010, followed by the brightening from 2011 to 2015, and then by a dimming from 2016 to 2017. Howbeit the long-term trends of GSR is not directly related to the variations of DIF in corresponding regions. For instance, neither obvious brightening nor dimming is 15 manifested in the northwest while the dimming of DIF in the Qinghai-Tibet Plateau continues up to 2015. In recent years (2016 - 2018), dramatic variations of solar radiation are observed in summer.

# 4 Validation of the radiation datasets

## 4.1 Overall performance

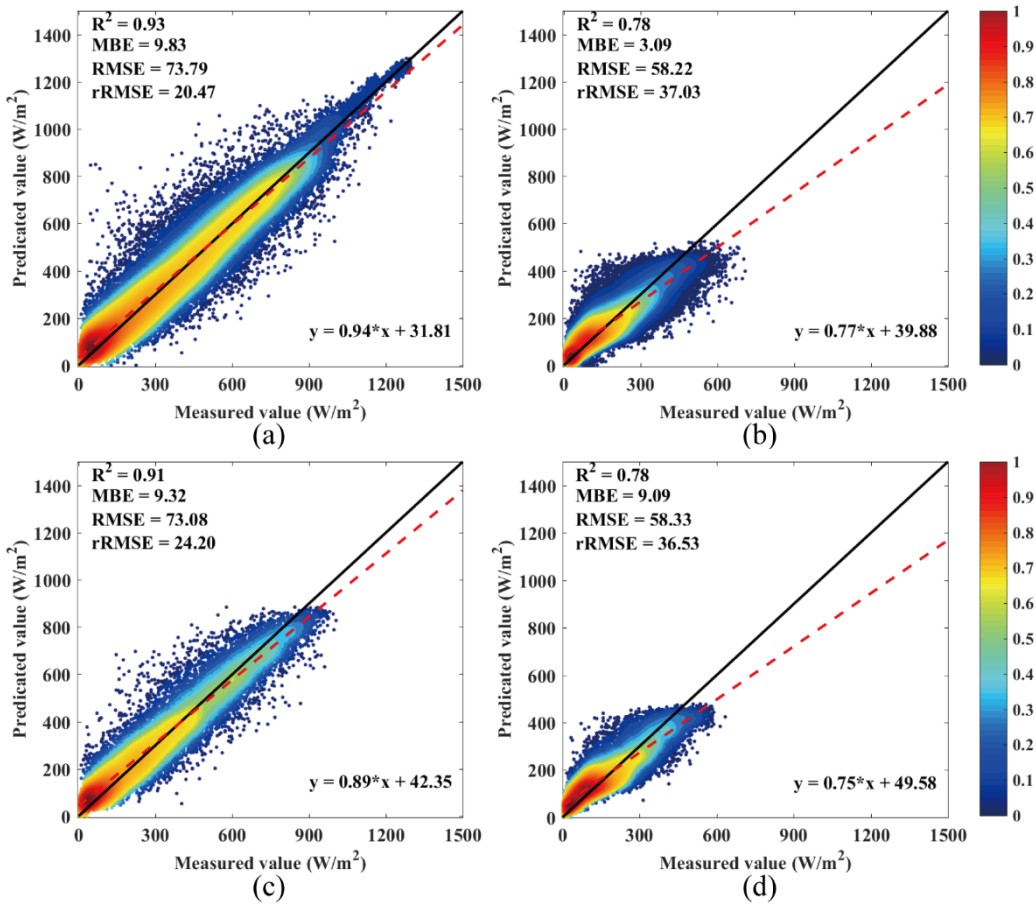

**Figure 6: The density plots of predicted hourly GSR (first column) and DIF (second column) versus measured values at (a-b) all sites and the top of Mt. Everest and (c-d) 5 independent validation sites in 2008. The black solid lines are 1:1 lines and the red dashed lines represent the fitted regression lines, whose expressions are labelled at the lower right corner.**

The performance of the hybrid network is evaluated by means of comparing predicted values with ground measurements as shown in figure 6. Overall, it provides satisfactory estimates for hourly radiation at the site scale with an $R^2$ of 0.93, MBE of 9.83 W/m$^2$, RMSE of 73.79 W/m$^2$ and rRMSE of 20.47 for GSR, whereas an $R^2$ of 0.78, MBE of 3.09 W/m$^2$, RMSE of 58.22 W/m$^2$ and rRMSE of 37.03 for DIF, as shown in figure 6a-b. The positive MBE values confirm that our production overestimates the surface radiation at some degree, which might attribute to the relative lower measurements due to the urbanization effects (Wang et al., 2014). The errors of DIF are more serious than GSR indicated by the lower $R^2$ and larger rRMSE. The potential causes are the higher requirements during DIF estimation for explicit representation of aerosols, clouds, and their interactions. With respect to the five independent sites (figure 6c-d), it achieves an $R^2$ of 0.91, MBE of 9.32



W/m$^2$, RMSE of 73.08 W/m$^2$ and rRMSE of 24.20 for GSR, and an R$^2$ of 0.78, MBE of 9.09 W/m$^2$, RMSE of 58.33 W/m$^2$ and rRMSE of 36.53 for DIF, proving that our hybrid network owns perfect extendibility in space. Still, the performance on DIF is slightly worse than that on GSR. Anyhow, the above RMSE values are comparable to those in existing models, for instance, Shamim et al. (2015) achieved a testing RMSE of GSR about 110.83 W/m$^2$; the RMSE values of GSR ranged from 131.39 to 142.22 W/m$^2$ from Yao et al. (2015). Owing to the appropriate representation of changeable cloud morphology and non-linear relationships between inputs and outputs, the deep network provides the most exciting estimation of hourly radiation. The performance on GSR estimation at daily total and monthly-averaged daily total scales with an RMSE of 2.14 MJ/m$^2$ and 1.30 MJ/m$^2$, respectively, is also remarkably better than previous studies. For example, the testing RMSE of daily total GSR was 3.09 MJ/m$^2$ from Lu et al. (2011) and 2.93 MJ/m$^2$ from Landeras et al. (2012); the RMSE of monthly-averaged daily total GSR for validation was 3.94 MJ/m$^2$ from Senkal and Kuleli (2009) and 1.92 MJ/m2 from Lu et al. (2011).

## 4.2 Temporal distribution of errors

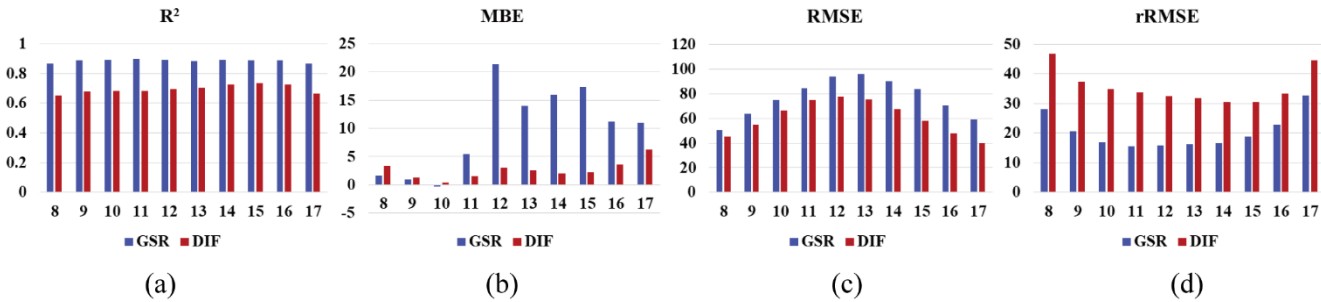

**Figure 7: The R$^2$, MBE, RMSE and rRMSE between predicted hourly GSR/DIF and ground measurements in 2008 grouped by the local hour from 8:00 to 17:00. The units of MBE and RMSE are W/m$^2$.**

Figure 7a-d shows the total errors of hourly predictions grouped by the local hour from 8:00 to 17:00. All predictions correlate well with the ground measurements with the lowest R$^2$ being 0.86 for GSR and 0.65 for DIF, fully proving the superior performance of deep network in hourly radiation estimation again. Large MBEs are likely to occur in the afternoon due to the high intensity of solar radiation. Although the MBE and RMSE illustrate that errors at hours near noon are much larger along with the increasing intensity of solar radiation, the performances are still promising given that rRMSE are all below 35%. Because the amounts of received surface radiation in the morning and at night are originally very low, rRMSE values at related hours are conversely large regardless of the relatively small RMSE. In the temporal dimension, these deviations might result from the difference that the satellite images reflect an instantaneous state of the atmosphere whereas ground measurements represent the average state within per unit time (herein one hour). If the clouds move rapidly, the ground stations would be covered by cloud shadows within a certain period but finally under a clear condition at the time the sensor scans so that the ground measurements are much smaller than satellite-based inversion values. Therefore, large

positive MBEs are likely to occur when coming across changeable clouds. As the deep network just takes into consideration the spatial concentration of solar radiation, it cannot properly handle the lag effect and cumulative effect in time series, thus is difficult to simulate the dramatic change in a short time. The recurrent neural networks (Heck and Salem, 2017; Hochreiter and Schmidhuber, 1997) which allow for exhibition of temporal dynamic behaviour, are the promising solutions. In addition,

both figure 7a and 7b confirm that the prediction accuracy of GSR surpasses that of DIF, suggesting that using only the visible information of geostationary imagery may not be enough for the deep networks to fully capture the sophisticated atmospheric scattering mechanism.

## 4.3 Spatial distribution of errors

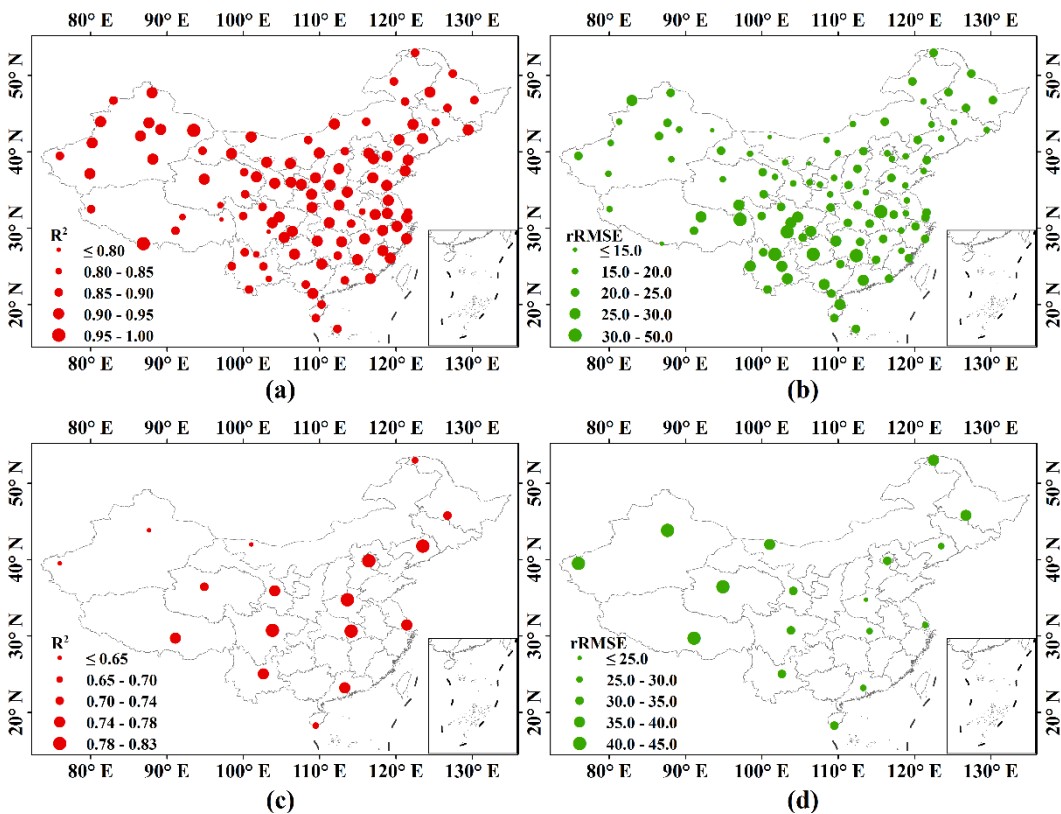

**Figure 8: The spatial distribution of $R^2$ and rRMSE in 2008 at each site. (a) $R^2$ for GSR; (b) rRMSE for GSR; (c) $R^2$ for DIF; (d) rRMSE for DIF. Note that DIF is measured only at 17 radiation sites.**

The $R^2$ and rRMSE of hourly GSR/DIF at each site are displayed in figure 8, where obvious geographical differentiation is observed. On the whole, estimates from our production correlate well with ground observations at sites with high probability of cloud-free skies, for instance, the north and northwest China. Low $R^2$ and large rRMSE are likely to occur at sites located

in regions with more cloudy days, such as the south and southwest China, especially the Szechwan Basin perennially





covered by clouds. It is particularly obvious that the correspondence is relatively worse from the Tibetan Plateau to southern China. The previous assessment on the ERA-Interim data in China (Zhang et al., 2018) suggests that the possible causes are various due to the spatially stratified heterogeneity of the atmosphere: the dominant factor is aerosols on the Tibetan Plateau, cloud coverage in the Sichuan Basin or on the Yunnan-Guizhou Plateau, and aerosols and water vapour content in the

Southeast Hills. In the spatial dimension, the errors might result from the fact that the gridded estimation from satellite data is inherently spatial domain-averaged while ground observations focus on the local concentration of solar radiation, easily affected by surrounding environments (Greuell and Roebeling, 2009). If the ground station is covered by shadows of cirrus clouds or surroundings, its real-time measured GSR will be lower than that of other locations within the matching pixel of satellite images, thus being lower than satellite-derived values. With regard to DIF, the correlation between predictions and

measurements is much worse than that of GSR, in agreement with the results in section 4.2. Contrary to the GSR, predications of DIF behave well in humid areas (southern China) rather than in arid areas (northwest China). It may attribute to that the deep network has only gained the knowledge of cloud depth, thus fails in accurately distinguishing the difference of scattering patterns between wet and dry climates due to the absence of information about atmospheric water vapour and particulate matter. Anyhow, the root cause still lies in the insufficiency of DIF training samples. From this point of view,

spatially uniformly distributed and densely installed surface radiation networks are in urgent requirements.

### 4.4 Comparisons to other products

To further verify the reliability of our datasets, we compare it to two widely-used radiation products: ERA5 reanalysis product from the European Centre for Medium-Range Weather Forecasts (ECMWF) (https://climate.copernicus.eu/ last accessed: 21 May 2019) and solar radiation from National Renewable Energy Laboratory (NREL) (https://www.nrel.gov/

last accessed: 21 May 2019). ERA5 is the fifth generation ECMWF atmospheric reanalysis of the global climate, providing estimates for each hour of the day on $0.25\,° \times 0.25\,°$ latitude-longitude grids. It adopts the Integrated Forecasting System (IFS Cycle 41r2), the perfect representative of the advanced research and development in modelling and data assimilation, to guarantee the significant increase in forecast accuracy, and then combines the model simulation with observations from across the world into a globally complete and consistent dataset using the laws of physics. NERL datasets are based on the

State University of New York/Albany satellite radiation model (Perez et al., 2002), developed at the National Renewable Energy Laboratory and other universities for the U.S. Department of Energy. The model uses hourly radiance images from geostationary weather satellites, daily snow cover data, and monthly averages of atmospheric water vapour, trace gases, and the amount of aerosols in the atmosphere to calculate the hourly total insolation (sun and sky) falling on a horizontal surface. Figure 9 shows the spatial distribution of yearly GSR, DIF and DIR from ECMWF ERA5 and NERL, which are comparable

to the results shown in figure 4g-i. Overall, our results are highly similar to that of NERL, but significantly differ from that of ERA5. Taking the NERL radiation as the reference, ERA5 underestimates DIF thoroughly except on the Tibetan Plateau and Pamirs Plateau and overestimates GSR and DIR in Taklimakan Desert and East China, while our estimates perform well



in most regions except the Tibetan Plateau and Pamirs Plateau where large differences on GSR/DIR are observed. That the derived DIF of ERA5 exhibits high values on the Tibetan Plateau (frequent clear skies) while low values in the Middle and Lower Reaches of the Yangtze River (frequent cloudy skies) is obviously contrary to our common knowledge. However, to validate the reliability of different products on the Tibetan Plateau additional observations are required. With respect to the neighbouring regions, although our estimates are erroneously deviated in India and Southeast Asia due to the lack of training samples, the relative differences of radiation caused by local cloud thickness are properly represented.

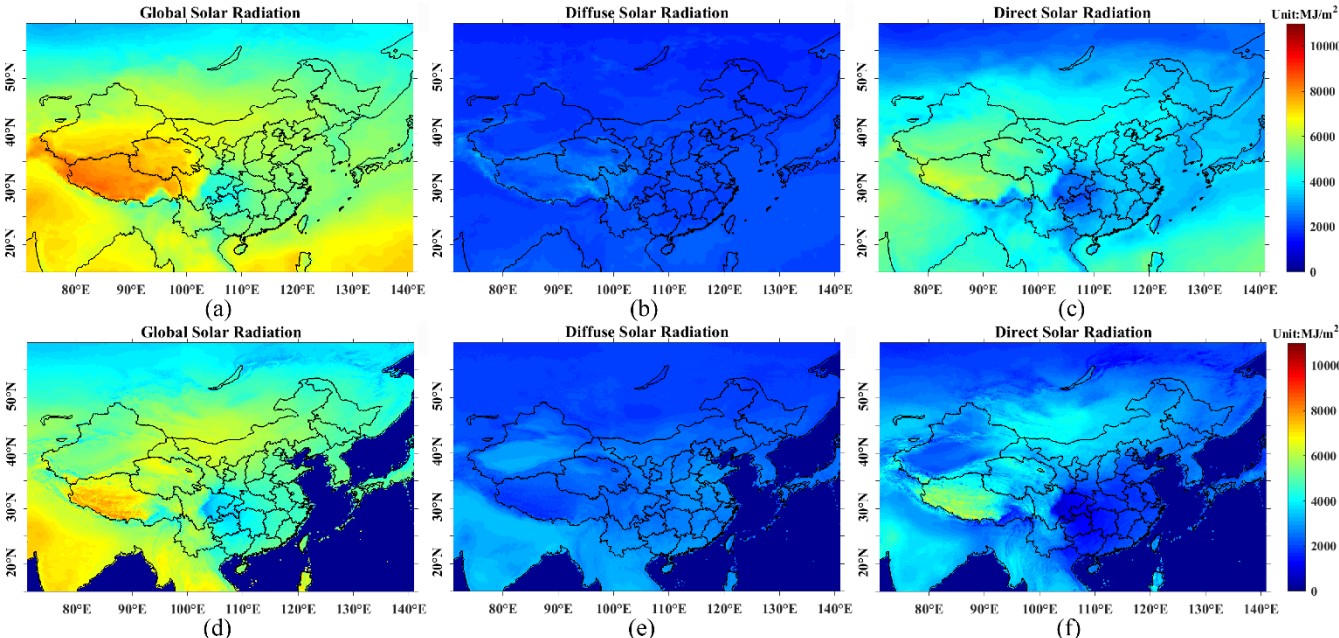

**Figure 9: Spatial distribution of yearly GSR, DIF and DIR from (a-c) ECMWF ERA5 and (d-f) NERL in MJ m$^{-2}$. Note that data from NERL are long-term yearly average of daily totals, covering the period from 1994/1999/2007 (depending on the region) to 2015 over the land area.**

**5 Data availability**

All hourly, daily and monthly radiation datasets from 2007 to 2018 are freely available from the Pangaea at https://doi.pangaea.de/10.1594/PANGAEA.904136 (Jiang and Lu, 2019), through which users can link to the specific data entities of each year. The dataset for one year includes twelve folders for hourly radiation, one folder for daily total radiation, one folder for monthly total radiation as well as other supporting documents. All radiation files are stored in HDF5 format and each file contains two variables representing (daily/monthly total) global radiation and (daily/monthly total) diffuse radiation, respectively. The files for hourly datasets, named as "RAD_yyyymmddhh.h5" where "yyyy", "mm", "dd", and "hh" denote year, month, day and hour (UTC time), are stored as int16 data type with an scaling factor of 0.01, whereas the



files for daily datasets (named as "RAD_yyyymmdd.h5" where "yyyy", "mm", and "dd" denote year, month, and day) and monthly datasets (named as "RAD_yyyymm.h5" where "yyyy" and "mm" denote year and month) are stored as floating-point data type with scaling factor of 1.00. The datasets provide gridded radiation data within 71.025 °E - 141.025 °E and 14.975 °N - 59.975 °N along with a spatial resolution of 0.05 ° in the unit of 0.01 MJ m$^{-2}$. The hourly radiation can also be expressed in unit of W/m$^2$ through the conversion: 0.01 MJ m$^{-2}$ hour$^{-1}$=1/0.36 W m$^{-2}$. The time coverage of hourly dataset is from 2007-01-01 0:00 to 2018-12-31 23:00 (UTC). More details and examples of data visualization can refer to the published description files in each dataset. It is noted that all hourly data are provided in UTC universal time.

## 6 Conclusions

Datasets on solar radiation is of great importance to the detection of global dimming and brightening; the quantification of the Earth's surface energy budget; the sustainable development of ecological environments or vegetation productivity, the simulation of regional climate models, and assessments of solar resources for solar power production. For the first time, we introduce a new satellite-based radiation inversion method which relies on CNN to characterize changeable cloud morphology of satellite imagery and MLP to build non-linear relationships between satellite signals and ground observations, to generate high-precision hourly radiation data series, including GSR and DIF over China from 2007 to 2018. The method achieves the integration of spatial pattern and the simulation of complex radiation transfer process, finally yielding an overall R$^2$ of 0.93, MBE of 9.83 W/m$^2$, RMSE of 73.79 W/m$^2$ and rRMSE of 20.47 for hourly GSR, and an R$^2$ of 0.78, MBE of 3.09 W/m$^2$, RMSE of 58.22 W/m$^2$ and rRMSE of 37.03 for hourly DIF. The RMSE of daily total and monthly-average daily total GSR decrease to 2.14 MJ/m$^2$ and 1.30 MJ/m$^2$, respectively, notably superior to the results of existing algorithms. The spatially continuous maps also properly reflect the temporal variations and spatial distribution in fine scales. Error analysis of our datasets and comparison to other products reveal that fusion of time series images along with auxiliary information from other channels could further improve the data accuracy and advance in reproducing the short-term fluctuation of solar radiation. Next, we intend to optimize the hybrid network based on above findings, and then to produce long-term time series solar radiation datasets for change detection and tendency analysis. We expect that our products would serve as a key data source on spatiotemporal distribution of solar radiation in support of related scientific researches and industrial applications in the future.





**Author contributions.** HJ and NL developed the deep network and generated the datasets, LY collected related data and performed data pre-processing. HJ wrote the manuscript. NL and LY provided assistance in the organization of this articles.

**Competing interests.** The authors declare that they have no conflict of interest.

**Acknowledgements.** We are very grateful to the China Meteorological Administration for providing the surface radiation observation data. The MTSAT satellite data were obtained from the Kochi University and SRTM DEM data were available from the U.S. Geological Survey. This work was supported by the National Natural Science Foundation of China (No.41971312, 41890854).

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
