# Peer review of "Surface global and diffuse solar radiation over China acquired from geostationary Multi-functional Transport Satellite data"

_Earth System Science Data, 2019_

## Referee Comment (RC1) · Anonymous Referee #1 · 6 Dec 2019

Summary and comments

Jiang et al. produced total and diffuse surface radiation over China using the geostationary satellite data and the deep learning network. I find the dataset of general interest. I have some concerns related to this manuscript.

(1) the title suggests it is surface global solar radiation over China. I think "global" here should represent the whole earth. The title needs to be changed.

(2) scientific relevance: I do not think the authors provide enough details about how the new technique can address previous research problems/scientific questions. For example, what do the authors mean "no consideration of spatial collocation of surface

radiation", and how the proposed deep learning algorithm addresses this consideration remains unclear. What spatial patterns have been extracted from satellite images and how they are linked to target hourly radiation values are also not clear. The introduction gives an impression that the dataset is probably "over-produced" with no clear clue that how these datasets will be used specifically. Although they first paragraph in the introduction mentioned some possible applications of these datasets, further digging of these literatures would tell you the produced dataset here would be useless. I would suggest the authors further clearly identify, for example, how diffuse radiation are being used to forecast crop yield (if there is a crop model uses it, please name it), and how diffuse radiation are being used to simulate carbon dynamics (if there are Earth system models requiring these datasets, please list them in the paper) or any other specific applications that use this type of datasets either partly or over the whole part of China (rather than talk them general). Based on current descriptions, I think the produced datasets would be useless.

(3) Methods and Comparisons with other products: I think throughout the manuscript, there is no scientific explanation that why we should correlate satellite images of five bands with total/diffuse solar radiation (I think the author also need to provide a definition in this manuscript what exactly data they are providing). I believe what you mean solar radiation here should be the integration of radiation over the whole wavelength rather just a few wavelengths. Although the authors use the deep learning algorithm, it is necessary to explain what the mechanisms behind this correlation. Particularly, as the authors mentioned previously that spatial patterns are being extracted and correlated with target data from point locations, the authors should also explain what patterns are being used. Without sufficient explanations, the readers may concern about what are the error estimates and sources which the authors do not provide at all. For example, how reliable can we use the solar radiation estimates under cloudy conditions? Can the authors provide a quality flag for these areas and the confidence interval for us to use the dataset for regions under cloudy conditions? Does the authors separate comparisons/validation in accuracy for point locations between clear-sky and

cloudy conditions? What is the accuracy level under clear-sky conditions and what is the accuracy level under cloudy conditions? IF the comparison in accuracy between this produced product and the other product under clear-sky conditions is the same or similar, what are the advantages of this produced product compared to previous ones? In addition, for the deep learning part, I do not understand why we use 16*16 pixels. How you get the size determined? Plus, do the radiation measurements from point locations are across all these 16 pixels or just belong to only one of them? I have the impression that because the authors would like to use deep learning, they need to have an image used as input. So, this is a point to area comparison? If so, I would like to see the authors providing error estimates related to this misrepresentation of points as areas for the produced datasets.

(4) My last concern is that I think the manuscript has duplication issues: some of the figures have been seen previously in your other manuscripts. for example, figure 1, 2, etc (in your recent publications in Renewable and Sustainable Energy Reviews, Volume 114, October 2019, 109327). Also, I would like to mention here that previously you used the ResnetTL, and in this manuscript, it seems that you use a different network structure. So, what is the difference between them? what are the improvements?

The authors also need to pay attention to the grammar of the manuscript and the language needs to be further edited.

---

## Referee Comment (RC2) · Anonymous Referee #2 · 15 Dec 2019

Hou et al. present a study about "surface global and diffuse solar radiation over China acquired from geostationary Multi-functional Transport Satellite data". The following questions should be satisfactorily answered before consideration for publication: 1. The topic is not innovative enough, which has been done by many researchers, for example Tang et al (2016) has published an article named "Retrieving high-resolution surface solar radiation with cloud parameters derived by combining MODIS and MTSAT data". The input data in your model, the spatial and temporal resolution of your output GSR values are similar to that in Tang' study. Only using a artificial intelligence model could not be an innovation idea. I would strongly advise the author(s) of this paper to rewrite their introduction section to give more explanation of the research background.

[Figure]

A very general sentence is not enough to demonstrate the research significance. 2. In Tang's study (Ditto), Tang et al., innovatively correlated the cloud optical properties to the satellite data that were used in your study. The main radiation dumping processes including Rayleigh scattering, aerosol extinction, ozone absorption, water vapor absorption, permanent gas absorption, and cloud extinction are considered in Tang's study. What is the scientific correlation between satellite signals in five bands with GSR and DIF values in your study? Author should explain the mechanisms in your model. 3. Please check the unit of GSR, DIF in Figure 4. The unit are different throughout your article. 4. Check the label of the color-bar in Figure 6. 5. Many statistical indicators (RMSE, MAE, rRMSE, R2) are used to evaluate the model accuracy. How to evaluate the overall model performance of your model? 6. On the 14th page of your article, you noted "on the whole, estimates from our production correlate well with ground observations at sites with high probability of cloud-free skies". As well known, the northern China and northwestern China are the area with the highest of dust aerosol particles in China, especially in summer. How do you detect clear-sky? Author should evaluate the model accuracy in clear-skies and cloudy skies, otherwise author could not get this conclusion above. As well known, the southern and southeastern China are the areas with abundant precipitable water vapor and dense cloud, which would strongly affect the accuracy of your model. How do explain the accuracy of the estimated DIF are higher in cloud weather conditions? Further sufficient explanation should be given for these questions. 7. Syntax check in the whole manuscript should be done. 8. The main contents of this article have been published previously in another journal. This is a serious academic moral issue. This article is highly repetitive with your previous articles on Renewable and sustainable Energy Reviews (https://doi.org/10.1016/j.rser.2019.109327). The Figure 1, Figure 2 have been used in your previously published article. Even the main method (CNN and MLP) and the main framework of this article are the same as that in previously published article.

In all, we think that this article is not prepared and should be rejected for publication on ESSD.

---

## Author Comment (AC1) · 15 Dec 2019

Response to Referee #1

We would like to thank the reviewer for the comments and suggestions, which are all valuable and very helpful for improving our paper. We have made revisions and a point-to-point response is present in the following. Pleas find the attached supplement file for a clear layout and responses with colors.

Summary and comments: Jiang et al. produced total and diffuse surface radiation over China using the geostationary satellite data and the deep learning network. I find the

dataset of general interest. I have some concerns related to this manuscript.

Response: We thank Referee #1 for the encouraging comments. All comments and suggestions have been considered carefully and well addressed.

Comments: (1) the title suggests it is surface global solar radiation over China. I think "global" here should represent the whole earth. The title needs to be changed.

Response: Thank you for your advice. In the revised manuscript, the title has been changed to "Surface solar radiation and the diffuse component in China acquired from geostationary Multi-functional Transport Satellite data"

Comments: (2) scientific relevance: I do not think the authors provide enough details about how the new technique can address previous research problems/scientific questions. For example, what do the authors mean "no consideration of spatial collocation of surface radiation", and how the proposed deep learning algorithm addresses this consideration remains unclear. What spatial patterns have been extracted from satellite images and how they are linked to target hourly radiation values are also not clear.

Response: Thank you for your advice. The expression of "no consideration of spatial collocation of surface radiation" might be incorrect and confusing. In fact, we meant that "takes no consideration of spatial adjacent effects of surface radiation". In general, the proposed deep learning algorithm addresses the spatial adjacent effects by dealing with spatial satellite image blocks of 16×16 pixels. Hierarchical features from low-level details (e.g., geometric shapes, sizes, orientations, edges and distribution) to high-level comprehensive abstract representations (e.g., intrinsic physical and optical properties of mixed clouds) are extracted from satellite images to handle the spatial adjacent effects. Then, the multi-layer perceptron is utilized to link extracted features to target hourly radiation values through implicit non-linear expressions, and their parameters are learnt from pre-prepared training samples in supervised manner. The details are described in the Introduction of the revised manuscript. The related parts are as "These methods are in theory based on an independent pixel approximation

which assumes a plane-parallel horizontally homogeneous cloud. Thus, surface radiation retrievals from satellite imagers are pixel-based (from point to point), in other words, only (multi-band) satellite signals corresponding to the specific ground location are used for surface radiation estimation. However, in reality this idealized situation does not always exist, or even is uncommon. For example, in the presence of broken clouds, multiple reflections and scattering events off the sides of clouds or on the surface would lead to significant horizontal photon transport (Madhavan et al. 2017; Oreopoulos et al. 2000; Schewski and Macke 2003), which makes significant differences when the spatial resolution increases to several kilometres where the surface radiation of an individual footprint under inhomogeneous clouds is relevant to multiple adjacent satellite pixels (Huang et al. 2019). In fact, large biases and uncertainties occur frequently under broken clouds when comparing current high-resolution surface radiation products to quality-controlled ground observations (Deneke et al. 2009; Huang et al. 2016)." Therefore, it seems that area-to-point retrievals are the optimal solutions. From this point of view, a practical effort has been made in our previous works, where a hybrid deep network mainly consisting of convolutional neural network (CNN) blocks and multi-layer perceptron (MLP) is built to retrieve hourly GSR/DIF from geostationary satellite data (Jiang et al. 2019). The CNN blocks takes image blocks as inputs thereby allowing for identical treatment of adjacent satellite pixels, and are further stacked to construct deep residual structure to extract hierarchical features from low-level details (e.g., geometric shapes, sizes, orientations, edges and distribution) to high-level comprehensive abstract representations (e.g., intrinsic physical and optical properties of mixed clouds). It is believed that such hierarchical architecture of spatial features can fully expose the scattering effects, absorption effects as well as their interactions in the atmosphere, thus can be considered as substitutes for various input parameters representing atmospheric state in radiative transfer models. The MLP is utilized to link extracted features of CNN and additional auxiliary information defining the state in time and space to target measurements of hourly surface radiation through implicit non-linear expressions, whose parameters are learnt from pre-prepared training samples
in supervised manner. The deep network is demonstrated to be effective in handling spatial adjacent effects and simulating complicated radiative transfer processes, and successful in achieving superior accuracy of GSR estimates."

Comments: The introduction gives an impression that the dataset is probably "over-produced" with no clear clue that how these datasets will be used specifically. Although they first paragraph in the introduction mentioned some possible applications of these datasets, further digging of these literatures would tell you the produced dataset here would be useless. I would suggest the authors further clearly identify, for example, how diffuse radiation are being used to forecast crop yield (if there is a crop model uses it, please name it), and how diffuse radiation are being used to simulate carbon dynamics (if there are Earth system models requiring these datasets, please list them in the paper) or any other specific applications that use this type of datasets either partly or over the whole part of China (rather than talk them general). Based on current descriptions, I think the produced datasets would be useless.

Response: Thank you for your advice. More specific models (e.g., JULES, FöBAAR, YIBs, SWAP) and applications (e.g., modelling radiation-use efficiency of wheat, early yield assessment of soybean, wheat and sunflower) have been added into the first paragraph of the revised Introduction. The related parts are as follows: "The explicit knowledge on DIF is urgently required to assess its effects on plant productivity and carbon dynamics of terrestrial ecosystems that has become a popular issue in the field of ecology and environmental sciences (Gu et al. 2002; Mercado et al. 2009; Zhang et al. 2011; Zhang et al. 2017). For instance, downward direct and diffuse radiation at the surface are necessary inputs for JULES land-surface scheme and canopy radiation-photosynthesis scheme to account for effects of diffuse radiation on sunlit and shaded photosynthesis thus impact on global primary production (Mercado et al. 2009) and land carbon sink (Rap et al. 2018), and also used to calculate photosynthetic photon flux density in the Forest Biomass, Assimilation, Allocation, and Respiration (FöBAAR) model for the purpose of simulating forest carbon cycle (Lee et al. 2018). The perturbations in diffuse solar radiation are needed to be quantified when using the Yale Interactive terrestrial Biosphere (YIBs) model to estimate the response of global carbon cycle to fire pollutions (Yue and Unger 2018). It also proves that satellite-based radiation data improve the performance of Soil Water Atmosphere Plant (SWAP) model during crop yield estimation (Mokhtari et al. 2018). Besides, the partitioning of diffuse and direct solar radiation as well as their diurnal variations are essential for modelling radiation-use efficiency of wheat during its vegetative phase (Choudhury 2000) and the early assessment of crop (i.e., soybean, wheat and sunflower) yield on a daily or shorter basis (Holzman et al. 2018)."

Comments: (3) Methods and Comparisons with other products: I think throughout the manuscript, there is no scientific explanation that why we should correlate satellite images of five bands with total/diffuse solar radiation (I think the author also need to provide a definition in this manuscript what exactly data they are providing). I believe what you mean solar radiation here should be the integration of radiation over the whole wavelength rather just a few wavelengths.

Response: Thank you for your advice. In fact, we directly link the satellite signals of visible channel to target shortwave solar radiation provided by ground measurements, without integration of radiation over different wavelengths. The utilization of other channels such as IR1-4 might add useful information about water vapor, cloud temperature etc. for final radiation estimation, but to assure the cross-sensor applications, only the visible channel of MTSAT data are used as it is available for almost all satellite images. We have made it clear in the revised manuscript. The explanation is given in Section 2.1 as "The original MTSAT-1R satellite images are resampled to so-called hourly GAME products with a resolution of 1/20°, which is freely accessible at http://weather.is.kochi-u.ac.jp/ (last accessed: 10 Dec., 2019). Herein, only the visible channel is used for the inference of shortwave or diffuse solar radiation provided by ground measurements." and also in Section 2.2 as "In addition, for the convenience of cross-sensor applications, it is better to only depend on the visible channel which

is available for nearly all satellite images. This is reasonable as the visible channel provides the most proportion of information on aerosols, clouds and other atmospheric properties (Lu et al. 2011)."

Comments: Although the authors use the deep learning algorithm, it is necessary to explain what the mechanisms behind this correlation. Particularly, as the authors mentioned previously that spatial patterns are being extracted and correlated with target data from point locations, the authors should also explain what patterns are being used. Without sufficient explanations, the readers may concern about what are the error estimates and sources which the authors do not provide at all. For example, how reliable can we use the solar radiation estimates under cloudy conditions? Can the authors provide a quality flag for these areas and the confidence interval for us to use the dataset for regions under cloudy conditions? Does the authors separate comparisons/validation in accuracy for point locations between clear-sky and cloudy conditions? What is the accuracy level under clear-sky conditions and what is the accuracy level under cloudy conditions? IF the comparison in accuracy between this produced product and the other product under clear-sky conditions is the same or similar, what are the advantages of this produced product compared to previous ones?

Response: Thank you for your advice. In the revised manuscript, we have explained the potential mechanisms in the Introduction and Section 2.2. More explanations are added into the first paragraph of Section 2.2. The related parts in the Introduction are as follows: "The CNN blocks takes image blocks as inputs thereby allowing for identical treatment of adjacent satellite pixels, and are further stacked to construct deep residual structure to extract hierarchical features from low-level details (e.g., geometric shapes, sizes, orientations, edges and distribution) to high-level comprehensive abstract representations (e.g., intrinsic physical and optical properties of mixed clouds). It is believed that such hierarchical architecture of spatial features can fully expose the scattering effects, absorption effects as well as their interactions in the atmosphere, thus can be considered as substitutes for various input parameters representing atmospheric state

in radiative transfer models." The related parts in Section 2.2 are as follows: "Satellite image is regarded as a vivid portrayal of the atmosphere and the surface state, and its recorded signals usually contain information on cloud-radiation interactions and impacts among adjacent locations. Traditional physical algorithms retrieve surface radiation from satellite signals on the basis of various radiative transfer models or their simplified versions, where geometric conditions, atmospheric conditions, and aerosol types should be strictly defined, complex processes such as atmospheric absorption and scattering, and their interactions are needed to be precisely simulated, or clear-sky and cloudy retrieval modes are independently developed. Herein, we utilize deep learning technique to directly build the implicit correlations between satellite signals and surface radiation in view of its powerful approximation ability of continuous mapping function. Except that all-sky situations are under a unified framework and tedious intermediate simulations are avoided, different from classical pixel-based retrievals, the CNN blocks are able to deal with spatial adjacent effects of surface radiation, that is, the influence of neighboring pixels on the central point can be taken into account." Although the spatially continuous error estimates are not provided, the validation results at site level shown in figure 8 can be used as a reference for the rational utilization of our datasets. Different from the previous parameterization schemes, we didn't develop independent clear-sky and cloudy retrieval modes separately. The deep network estimates solar radiation under all-sky conditions in a unified manner, and provides reliable results as indicated by Figure 6. As labels indicating a clear-sky or cloudy condition are unavailable for hourly measurements, we didn't carry out separate comparisons during validation. In view of the high precision of our products under all-sky conditions, we suggest users in various fields to use it at ease, but limited within China

Comments: In addition, for the deep learning part, I do not understand why we use 16*16 pixels. How you get the size determined? Plus, do the radiation measurements from point locations are across all these 16 pixels or just belong to only one of them? I have the impression that because the authors would like to use deep learning, they need to have an image used as input. So, this is a point to area comparison? If so, I

would like to see the authors providing error estimates related to this misrepresentation of points as areas for the produced datasets.

Response: Thank you very much for your advice. Actually, the CNN blocks deal with image blocks of 16*16 pixels to infer the surface radiation at the location corresponding to the central point of the input image block. Maybe we can call it area-to-point retrieval. We know that there exists a certain spatial scale for ground observations. Multiple reflections and scattering events off the sides of clouds or on the surface lead to significant horizontal photon transport, so that adjacent pixels also influence the measured radiation on the ground. Therefore, it is reasonable to take neighboring pixels into account during radiation estimation of the central location. Here, we expect the CNN blocks can approximate the spatial adjacent effects thereby improving the final accuracy of radiation estimation. We have made it clear in Section 2.2 as "The structure is shown in figure 1b and the detailed configurations are listed in table 1. There are two input pipes: Input1 for MTSAT image blocks and Input2 for additional attributes including the local time (month, day and hour) and location (longitude, latitude and altitude) corresponding to the central point of Input1. The Output can be either GSR or DIF associated with the central point of Input1." and also in the Introduction as "For example, in the presence of broken clouds, multiple reflections and scattering events off the sides of clouds or on the surface would lead to significant horizontal photon transport (Madhavan et al. 2017; Oreopoulos et al. 2000; Schewski and Macke 2003), which makes significant differences when the spatial resolution increases to several kilometres where the surface radiation of an individual footprint under inhomogeneous clouds is relevant to multiple adjacent satellite pixels (Huang et al. 2019). In fact, large biases and uncertainties occur frequently under broken clouds when comparing current high-resolution surface radiation products to quality-controlled ground observations (Deneke et al. 2009; Huang et al. 2016). Therefore, it seems that area-to-point retrievals are the optimal solutions. From this point of view, a practical effort has been made in our previous works, where a hybrid deep network mainly consisting of convolutional neural network (CNN) blocks and multi-layer perceptron (MLP) is built
to retrieve hourly GSR/DIF from geostationary satellite data (Jiang et al. 2019). The CNN blocks takes image blocks as inputs thereby allowing for identical treatment of adjacent satellite pixels, and are further stacked to construct deep residual structure to extract hierarchical features from low-level details (e.g., geometric shapes, sizes, orientations, edges and distribution) to high-level comprehensive abstract representations (e.g., intrinsic physical and optical properties of mixed clouds)." The previous studies revealed that time series of central point and its neighbouring pixels are most correlative within an extent of approximately 60km for hourly GSR, so the direct size of image block should be 12 *12. But it does not fit in with the classical size of CNN networks. So we choose the size of 16*16, thus extra pixels are included which are also helpful for extraction of edge features. More descriptions are added into Section 2.2 in the revised manuscript as "Considering the recommendation that time series of central point and its neighbouring pixels are most correlative within an extent of approximately 60km for hourly GSR (Wyser et al. 2005; Deneke et al. 2009), the input size for CNN is designed as $16 \times 16$ pixels ($\sim$ 80km $\times$ 80km) around the central location requiring for radiation estimation, slightly larger than recommended 60km to fit in with the classical CNN network structure and meanwhile ensure the extraction of edge features. In addition, for the convenience of cross-sensor applications, it is better to only depend on the visible channel which is available for nearly all satellite images. This is reasonable as the visible channel provides the most proportion of information on aerosols, clouds and other atmospheric properties (Lu et al. 2011)."

Comments: (4) My last concern is that I think the manuscript has duplication issues: some of the figures have been seen previously in your other manuscripts. for example, figure 1, 2, etc (in your recent publications in Renewable and Sustainable Energy Reviews, Volume 114, October 2019, 109327). Also, I would like to mention here that previously you used the ResnetTL, and in this manuscript, it seems that you use a different network structure. So, what is the difference between them? what are the improvements?

Response: Thank you very much for your advice. Actually, the structure of the network in this manuscript is similar to the ResnetTL. In this paper, we extend the ResnetTL to fit the estimation of diffuse radiation through transfer learning, an approach to reuse already gained knowledge to solve different but analogous problems. Therefore, a new deep network for DIF estimation is obtained by fine-tuning ResnetTL using new training samples consisting of ground measured diffuse radiation and the corresponding satellite image block. Both the trained ResnetTL and the network for diffuse radiation estimation are used together to generate our datasets. More explanations are added into the revised manuscript as in the Introduction "In this paper, we extend the previous network for GSR to fit the estimation of diffuse radiation through transfer learning, an approach to reuse already gained knowledge to solve different but analogous problems. A new deep network for DIF estimation is obtained by fine-tuning the GSR network using new training samples consisting of ground measured diffuse radiation and the corresponding satellite image block. After complete learning and optimization, the trained DIF network in combination with previous GSR network is used to generate radiation datasets (GSR and DIF) over China based on Multi-functional Transport Satellites (MTSAT) data.", and also in Section 2.2 as "In the previous work (Jiang et al. 2019), we have built a hybrid deep network for GSR estimation. Herein, we further optimize the GSR model to fit the estimation of diffuse radiation by fine-tuning (refer to Section 2.3). The structure is shown in figure 1b and the detailed configurations are listed in table 1." The used radiation stations are all the same to that in our previous paper. To avoid the duplication issues, we have revised related figures, for example, the background of figure 2 is changed as land cover types that is a perfect way to demonstrate the representativeness of our stations, the graphical structure of the deep network (figure 1b) is expressed in another way. These changes are added into the revised manuscript.

Comments: The authors also need to pay attention to the grammar of the manuscript and the language needs to be further edited.
Response: Thank you very much for your advice. We have checked the grammar throughout the manuscript and corrected related errors.

Please also note the supplement to this comment:
https://www.earth-syst-sci-data-discuss.net/essd-2019-209/essd-2019-209-AC1-supplement.pdf

————————————————————

---

## Author Comment (AC2) · 18 Dec 2019

**Response to Referee #2**

We would like to thank the reviewer for the comments and suggestions, which are all valuable and very helpful for improving our paper. We have made revisions and a pointto-point response is present in the following.

**Comments:**

Hou et al. present a study about "surface global and diffuse solar radiation over China acquired from geostationary Multi-functional Transport Satellite data". The following questions should be satisfactorily answered before consideration for publication:

1. The topic is not innovative enough, which has been done by many researchers, for example Tang et al (2016) has published an article named "Retrieving high-resolution surface solar radiation with cloud parameters derived by combining MODIS and MTSAT data". The input data in your model, the spatial and temporal resolution of your output GSR values are similar to that in Tang' study. Only using a artificial intelligence model could not be an innovation idea. I would strongly advise the author(s) of this paper to rewrite their introduction section to give more explanation of the research background. A very general sentence is not enough to demonstrate the research significance.

**Response:**

Thank you for your advice.

Yes, the topic of radiation is not new and has been done by many researchers in view of its general interests in many fields and supports to various basic models (e.g., JULES, FöBAAR, YIBs, SWAP) and applications (e.g., modelling radiation-use efficiency of wheat, early yield assessment of soybean, wheat and sunflower), as described in the Introduction of our revised manuscript.

In theory, previous studies (including Tang et al., 2016) are based on an independent pixel approximation which assumes a plane-parallel horizontally homogeneous cloud. Thus, their radiation retrievals are pixel-based (from point to point), in other words, only (multi-band) satellite signals or multi-source information corresponding to the specific ground location is used for surface radiation estimation. However, it has been demonstrated that multiple reflections and scattering events off the sides of clouds or on the surface would lead to significant horizontal photon transport, so that adjacent pixels within a certain spatial extent also influence the measured radiation at a specific location

on the ground. To the best of our knowledge, the traditional pixel-based retrieval cannot handle such spatial adjacent effects of surface radiation and none of operational methods has been proposed currently. Thus our innovation idea is to deal with image blocks of 16\*16 pixels through CNN blocks to infer the surface radiation at the location corresponding to the central point of the input image block. We expect the CNN blocks can approximate the spatial adjacent effects thereby improving the final accuracy of radiation estimation. The CNN blocks allow for identical treatment of adjacent satellite pixels, and can be stacked to construct deep residual structure to extract hierarchical features from low-level details (e.g., geometric shapes, sizes, orientations, edges and distribution) to high-level comprehensive abstract representations (e.g., intrinsic physical and optical properties of mixed clouds). It is believed that such hierarchical architecture of spatial features can fully expose the scattering effects, absorption effects as well as their interactions in the atmosphere, thus can be considered as substitutes for various input parameters representing atmospheric state in radiative transfer models. Meanwhile, the MLP can be utilized to link extracted features of CNN and additional auxiliary information to target measurements of hourly surface radiation through implicit nonlinear expressions. More explanations have been added into the Introduction of revised manuscript as "These algorithms mainly include two categories: constructing empirical mathematical relationships between top of atmosphere (TOA) and surface radiative fluxes (Linares-Rodriguez et al. 2013; Lu et al. 2011) and driving complex radiative transfer models utilizing satellite-derived atmospheric parameters (Greuell et al. 2013; Huang et al. 2011). These methods are in theory based on an independent pixel approximation which assumes a plane-parallel horizontally homogeneous cloud. Thus, surface radiation retrievals from satellite imagers are pixel-based (from point to point), in other words, only (multi-band) satellite signals corresponding to the specific ground location are used for surface radiation estimation. However, in reality this idealized situation does not always exist, or even is uncommon. For example, in the presence of broken clouds, multiple reflections and scattering events off the sides of clouds or on the surface would lead to significant horizontal photon transport (Madhavan et al. 2017; Oreopoulos et al. 2000; Schewski and Macke 2003), which makes significant differences when the spatial resolution increases to several kilometres where the surface radiation of an individual footprint under inhomogeneous clouds is relevant to multiple adjacent satellite pixels (Huang et al. 2019). In fact, large biases and uncertainties occur frequently under broken clouds when comparing current high-resolution surface radiation products to quality-controlled ground observations (Deneke et al. 2009; Huang et al. 2016).

Therefore, it seems that area-to-point retrievals are the optimal solutions. From this point of view, a practical effort has been made in our previous works, where a hybrid deep network mainly consisting of convolutional neural network (CNN) blocks and multi-layer perceptron (MLP) is built to retrieve hourly GSR/DIF from geostationary satellite data (Jiang et al. 2019). The CNN blocks takes image blocks as inputs thereby allowing for identical treatment of adjacent satellite pixels, and are further stacked to construct deep residual structure to extract hierarchical features from low-level details (e.g., geometric shapes, sizes, orientations, edges and distribution) to high-level comprehensive abstract representations (e.g., intrinsic physical and optical properties of mixed clouds). It is believed that such hierarchical architecture of spatial features can fully expose the scattering effects, absorption effects as well as their interactions in the atmosphere, thus can be considered as substitutes for various input parameters representing atmospheric state in radiative transfer models. The MLP is utilized to link extracted features of CNN and additional auxiliary information defining the state in time and space to target measurements of hourly surface radiation through implicit non-linear expressions, whose parameters are learnt from pre-prepared training samples in supervised manner. The deep network is demonstrated to be effective in handling spatial adjacent effects and simulating complicated radiative transfer processes, and successful in achieving superior accuracy of GSR estimates."

Here, we would like to point out the differences between our works and Tang's:

1) The inputs of our algorithm involve only the visible band of MTSAT data along with additional attributes including the local time (month, day and hour) and location (longitude, latitude and altitude) without other cloud parameters.

2) The MTSAT data are input into the model as a whole to handle the spatial adjacent effects of surface radiation.

3) Much higher accuracy of radiation data has been achieved in our work as shown in Figure R1, which further demonstrate the importance to take spatial adjacent effects into consideration during satellite-based radiation retrievals.

Figure R1: The density plots of predicted hourly global solar radiation versus measured values for (a) Tang's data and (b) our datasets in 2007.

4) The diffuse radiation is also estimated and provided in our datasets, which is comparable to the widely-used ECMWF-ERA5 data as shown in R2.

Figure R2: The density plots of predicted hourly diffuse solar radiation versus measured values for (a) ECMWF ERA5 and (b) our datasets in 2007.

**Comments:**

2. In Tang's study (Ditto), Tang et al., innovatively correlated the cloud optical properties to the satellite data that were used in your study. The main radiation dumping processes including Rayleigh scattering, aerosol extinction, ozone absorption, water vapor

absorption, permanent gas absorption, and cloud extinction are considered in Tang's study. What is the scientific correlation between satellite signals in five bands with GSR and DIF values in your study? Author should explain the mechanisms in your model.

**Response:**

Thank you for your advice. I am so sorry for the ambiguous explanation of our model and its mechanisms.

Different from Tang's work, we directly correlate the ground measured GSR or DIF values to the visible channel of MTSAT satellite data, and we expect the deep network to build their relationships through hierarchical feature representations of CNN and continuous mapping function approximation of MLP by learning from pre-prepared training samples in a supervised manner. The radiation dumping processes such as Rayleigh scattering, aerosol extinction, ozone absorption, water vapor absorption, permanent gas absorption, and cloud extinction are avoided as it is believed that hierarchical architecture of spatial features can fully expose above-mentioned effects, thus can be considered as substitutes for various input parameters representing atmospheric state in radiative transfer models. The utilization of other channels such as IR1-4 of MTSAT might add useful information about water vapor, cloud temperature etc. for radiation estimation, but herein to assure the cross-sensor applications, only the visible channel of MTSAT data are used as it is available for almost all satellite images. This is reasonable as the visible channel provides the most proportion of information on aerosols, clouds and other atmospheric properties. More explanations on the mechanisms in our model are added into the Introduction and Section 2.2 of the revised manuscript. The related parts in the Introduction are as "The CNN blocks takes image blocks as inputs thereby allowing for identical treatment of adjacent satellite pixels, and are further stacked to construct deep residual structure to extract hierarchical features from low-level details (e.g., geometric shapes, sizes, orientations, edges and distribution) to high-level comprehensive abstract representations (e.g., intrinsic physical and optical properties of mixed clouds). It is believed that such hierarchical architecture of spatial features can fully expose the scattering effects, absorption effects as well as their interactions in the atmosphere, thus can be considered as substitutes for various input parameters representing atmospheric state in radiative transfer models. The MLP is utilized to link

extracted features of CNN and additional auxiliary information defining the state in time and space to target measurements of hourly surface radiation through implicit non-linear expressions, whose parameters are learnt from pre-prepared training samples in supervised manner. The deep network is demonstrated to be effective in handling spatial adjacent effects and simulating complicated radiative transfer processes, and successful in achieving superior accuracy of GSR estimates." The related parts in Section 2.2 are as "Satellite image is regarded as a vivid portrayal of the atmosphere and the surface state, and its recorded signals usually contain information on cloud-radiation interactions and impacts among adjacent locations. Traditional physical algorithms retrieve surface radiation from satellite signals on the basis of various radiative transfer models or their simplified versions, where geometric conditions, atmospheric conditions, and aerosol types should be strictly defined, complex processes such as atmospheric absorption and scattering, and their interactions are needed to be precisely simulated, or clear-sky and cloudy retrieval modes are independently developed. Herein, we utilize deep learning technique to directly build the implicit correlations between satellite signals and surface radiation in view of its powerful approximation ability of continuous mapping function. Except that all-sky situations are under a unified framework and tedious intermediate simulations are avoided, different from classical pixel-based retrievals, the CNN blocks are able to deal with spatial adjacent effects of surface radiation, that is, the influence of neighbouring pixels on the central point can be taken into account ... In addition, for the convenience of cross-sensor applications, it is better to only depend on the visible channel which is available for nearly all satellite images. This is reasonable as the visible channel provides the most proportion of information on aerosols, clouds and other atmospheric properties (Lu et al. 2011)"

**Comments:**

3. Please check the unit of GSR, DIF in Figure 4. The unit are different throughout your article.

**Response:**

Thank you for your advice. We have checked it carefully throughout our article. In this paper, two units (i.e.,  $W/m^2$  and  $MJ/m^2$ ) are used. In Section 3, all figures are shown in

the unit of  $MJ/m^2$  corresponding to the unit of the published datasets. To compare the spatial distribution of other products with our results in Figure 4, the figures in Figure 9 also shown in the unit of  $MJ/m^2$ . In contrast, the figure 6-8 during validation are displayed in the unit of  $W/m^2$  to keep identical with other studies of hourly radiation estimation (e.g., Tang et al., 2016) for the convenience of comparisons.

**Comments:**

4. Check the label of the color-bar in Figure 6.

**Response:**

Thank you for your advice. We have checked it carefully and added color-bar for each sub-plot as well as labels for each color-bar. More explanations about the density of color-bar are added in the figure title as "Figure 6: The density plots of predicted hourly GSR (first column) and DIF (second column) versus measured values at (a-b) all sites and the top of Mt. Everest and (c-d) 5 independent validation sites in 2008. The black solid lines are 1:1 lines and the red dashed lines represent the fitted regression lines, whose expressions are labelled at the lower right corner. Gaussian kernels are used for density plots and the density values are normalized to the 0–1 range through min-max normalization."

**Comments:**

5. Many statistical indicators (RMSE, MAE, rRMSE, R2) are used to evaluate the model accuracy. How to evaluate the overall model performance of your model?

**Response:**

Thank you for your advice. We have chosen four indicators, i.e.,  $R^2$ , MBE, RMSE, rRMSE, to evaluate the model accuracy from different aspects as any single index cannot demonstrate the comprehensive quality of datasets. In fact, it's almost impossible to develop a single indicator to assess the comprehensive model performance. For example, there are issues that the predicted hourly values are highly correlated to the measurements (large  $R^2$  values) but show significant overestimation or underestimation (large MBE values). In contrast, the utilization of multiple indicators could avoid such situation. The  $R^2$  measures the linear correlation between the observations and predictions, RMSE is

measure of their overall differences and MBE indicates whether predictions are overestimated or underestimated. For the convenience of comparison, rRMSE is also adopted as it is usually used in similar researches. The overall model performance of our model should be the comprehensive inspects of these four indicators. In addition, I wonder whether the word "overall" leads to misunderstanding. Herein, we use the word "overall" to mean that the model performance is validated on all measured records, not for different months or seasons, or for clear-sky and cloudy conditions. To avoid such misunderstanding, we have changed the title of Section 4.1 as "Validation against ground measurements"

**Comments:**

6. On the 14th page of your article, you noted "on the whole, estimates from our production correlate well with ground observations at sites with high probability of cloud-free skies". As well known, the northern China and northwestern China are the area with the highest of dust aerosol particles in China, especially in summer. How do you detect clear-sky? Author should evaluate the model accuracy in clear-skies and cloudy skies, otherwise author could not get this conclusion above. As well known, the southern and southeastern China are the areas with abundant precipitable water vapor and dense cloud, which would strongly affect the accuracy of your model. How do explain the accuracy of the estimated DIF are higher in cloud weather conditions? Further sufficient explanation should be given for these questions.

**Response:**

Thank you for your advice.

The conclusion that estimates from our production correlate well with ground observations at sites with high probability of cloud-free skies is incorrect. We have corrected as "On the whole, estimates from our production correlate well with ground observations at sites with low probability of cloudy conditions, for instance, the north and northwest China." Besides, more explanations are added as "Low R2 and large rRMSE are likely to occur at sites located in regions with more cloudy days, such as the south and southwest China, especially the Szechwan Basin perennially covered by clouds. As we know, both dust aerosol particles in the north and the northeast China and abundant

precipitable water vapor and dense cloud in the south and southeast China lead to nonclear skies, but the model performance is opposite in these areas. This might indicate that deep network does better in emulating the radiation effect of aerosols, but slightly worse in handling that of water vapor."

Different from the previous parameterization schemes, we didn't develop independent clear-sky and cloudy retrieval modes separately. The deep network estimates solar radiation under all-sky conditions in a unified manner, and provides reliable results as indicated by Figure 6. As labels indicating a clear-sky or cloudy condition are unavailable for hourly measurements, we didn't carry out separate comparisons during validation.

To investigate the causes for the contradiction that the accuracy of the estimated DIF are higher in cloudy weather conditions, we carry out an overfitting-test, in which the earlystopping mechanism is removed deliberately and the model for DIF estimate is trained repeatedly until it reaches an obvious over-fitting state where all training samples are intended to be well-fitted. If the model capability is responsible for the contradiction, changes in spatial distribution of  $R^2$  and rRMSE will be in our expectation. However, the results in figure R3-4 show that the model performance in the northwest China is improved but the spatial distribution keeps consistent with that in Figure 8c-d. This evidence points out that the low-quality diffuse measurements in the northwest China results in the apparently worse performance of diffuse estimation. In theory, on the premise that GSR model has proved its effectiveness in arid areas, the worse performance of DIF estimation under the same framework can only be attributed to the data quality. Further evidence also comes from the fact that measurements of diffuse radiation in the western China are not in a full-automatic tracking manner but involves manual operations, of which the nonstandard ones usually lead to measurement errors. More explanations are added into Section 4.3 in the revised manuscript as "With regard to DIF, the correlation between predictions and measurements is much worse than that of GSR, in agreement with the results in section 4.2. Contrary to the GSR, predications of DIF behave well in humid areas (southern China) rather than arid areas (northwest China), which is against our common sense that cloudy weather conditions in the southern China strongly affect the accuracy of radiation estimation. On the premise that GSR model has proved its effectiveness in arid areas, the worse performance of DIF estima

---

## Referee Comment (RC3) · Anonymous Referee #2 · 4 Jan 2020

Hou et al. have posted his response to Referee #2. However, we still think that this article should be rejected for publication. A manuscript or similar contents submitted on two magazines is not only seriously against the innovative principle of academic research, but also against scientific morale. This article is highly repetitive with your previous article published on "Renewable and sustainable energy reviews" (https://doi.org/10.1016/j.rser.2019.109327). Even some figures are copied from the article on RSER. Meanwhile, Tang et al. (2019) have published an article named "A 16-year dataset (2000–2015) of high-resolution (3‐h, 10‐km) global surface solar radiation" on ESSD. He generated a global solar radiation dataset with high accuracy, high spatial resolution and high temporal resolutions. What is your merit of your global

solar radiation data compared with that of Tang's? AI method could not explain the physical mechanism of the radiation dumping process on solar radiation. The diffuse solar radiation dataset is very valuable for solar researches and application. Thus, we suggested the author deleted the duplicated part of this article and resubmitted.

---

## Short Comment (SC1) · 9 Jan 2020

We would like to thank the reviewer for the comments and suggestions, which are all valuable and very helpful for improving our paper. We have made revisions and a point-to-point response is present in the following. Summary and comments: Hou et al. have posted his response to Referee #2. However, we still think that this article should be rejected for publication. A manuscript or similar contents submitted on two magazines is not only seriously against the innovative principle of academic research, but also against scientific morale. This article is highly repetitive with your previous article published on "Renewable and sustainable energy reviews"

[Figure]

(https://doi.org/10.1016/j.rser.2019.109327). Even some figures are copied from the article on RSER. Response: Thank you for your friendly comments. We are so sorry for the terrible writing and existing similarity on contents in the submitted draft. In fact, we are carrying out a systematic project on solar radiation involving algorithm, data and applications. The published article present our developed algorithm for surface solar radiation estimation with an original intention to improve radiation accuracy by deep learning techniques. Luckily, we have achieved such goal. Based on the previous work, we further extend the deep network for estimation of diffuse radiation and generate long-term data series. It is what this paper concentrate on. After your first round discussion, we have reorganized the paper, rewritten related parts and revised all figures in the paper. The updated manuscript focuses on the data itself with some new findings discussed, and all potential repetitiveness is avoided.

Comments: Meanwhile, Tang et al. (2019) have published an article named "A 16-year dataset (2000–2015) of high-resolution (3âAĽh, 10â ËŸ AĽkm) global surface solar radiation" on ESSD. He generated a global solar radiation dataset with high accuracy, high spatial resolution and high temporal resolutions. What is your merit of your global solar radiation data compared with that of Tang's? Response: Thank you for your advice. We think the merits of our datasets are that: 1) it provides more accurate estimation of global solar radiation at finer resolution of 5km which is important to reveal regional differences and changes of solar radiation; 2) the proportion of direct and diffuse is also available for our datasets with high accuracy. In the revised manuscript, we also compare our datasets to that of Tang's during validation.

Comments: AI method could not explain the physical mechanism of the radiation dumping process on solar radiation. Response: Thank you for your advice. Yes, it is difficult for deep learning to explain the definite physical mechanism behind its black-box operations. However, we think its mathematical principle is clear, i.e., implicitly construct the relationships between satellite signal and surface radiation through complex nonlinear expressions. We have also tried our best to explain its potential physical mechanism in

combination with related studies. More details and discussions on its mechanism are added into the revised manuscript.

Comments: The diffuse solar radiation dataset is very valuable for solar researches and application. Thus, we suggested the author deleted the duplicated part of this article and resubmitted. Response: Thank you very much for the encouraging suggestions. We will try our best to perfect the paper and improve our future work.

Please also note the supplement to this comment:
https://www.earth-syst-sci-data-discuss.net/essd-2019-209/essd-2019-209-SC1-supplement.pdf

———————————————————